# Agronomic and Technical Evaluation of Herbicide Spot Spraying in Maize Based on High-Resolution Aerial Weed Maps—An On-Farm Trial

**DOI:** 10.3390/plants13152164

**Published:** 2024-08-05

**Authors:** Alicia Allmendinger, Michael Spaeth, Marcus Saile, Gerassimos G. Peteinatos, Roland Gerhards

**Affiliations:** 1Department of Weed Science, Institute for Phytomedicine, University of Hohenheim, 70599 Stuttgart, Germany; alicia.allmendinger@uni-hohenheim.de (A.A.);; 2ELGO-DIMITRA, Leof Dimokratias 61, Agii Anargiri, 135 61 Athens, Greece; g.peteinatos@swri.gr

**Keywords:** spot spraying, weed detection, UAV imagery, weed control efficacy

## Abstract

Spot spraying can significantly reduce herbicide use while maintaining equal weed control efficacy as a broadcast application of herbicides. Several online spot-spraying systems have been developed, with sensors mounted on the sprayer or by recording the RTK-GNSS position of each crop seed. In this study, spot spraying was realized offline based on georeferenced unmanned aerial vehicle (UAV) images with high spatial resolution. Studies were conducted in four maize fields in Southwestern Germany in 2023. A randomized complete block design was used with seven treatments containing broadcast and spot applications of pre-emergence and post-emergence herbicides. Post-emergence herbicides were applied at 2–4-leaf and at 6–8-leaf stages of maize. Weed and crop density, weed control efficacy (WCE), crop losses, accuracy of weed classification in UAV images, herbicide savings and maize yield were measured and analyzed. On average, 94% of all weed plants were correctly identified in the UAV images with the automatic classifier. Spot-spraying achieved up to 86% WCE, which was equal to the broadcast herbicide treatment. Early spot spraying saved 47% of herbicides compared to the broadcast herbicide application. Maize yields in the spot-spraying plots were equal to the broadcast herbicide application plots. This study demonstrates that spot-spraying based on UAV weed maps is feasible and provides a significant reduction in herbicide use.

## 1. Introduction

Site-specific weed control offers great potential for herbicide savings due to the heterogeneous distribution of weeds in arable fields [1,2,3]. Patch spraying targets larger areas of aggregated weed populations. Mapping of weed patches, decision making and patch spraying was realized in three consecutive passes. Usually, GNSS-controlled sprayers with boom section valves have been used for patch spraying [4,5,6]. Patch spraying results in 40–60% herbicide savings without reducing weed control efficacy in arable fields [7]. Spot spraying involves targeting individual weed plants with a single nozzle [6,7]. It requires high-resolution real-time sensor technologies for weed detection and sprayers with single nozzle control.

Most commercial spot-spraying systems use sensors mounted on the sprayer. Some systems, e.g., Farming Revolution GT^®^ (Ludwigsburg, Germany), Ecorobotix ARA^®^ (Yverdon-les-Bains, Switzerland), Blue River See&Spray^®^ (Sunnyvale, CA, USA) and K.U.L.T.-inrow (Hamburg, Germany) apply artificial intelligence (AI)-based imaging software for weed/crop detection. However, most of these robots are restricted to certain crops because AI-based software was only trained for certain crops, weeding tools were designed only for certain crops or because the systems were designed for certain row spacing and crop architecture. Convolutional neural networks (CNNs)—artificial intelligence—are most commonly used to analyze sensor images. The driving speed of these robots is lower than for conventional sprayers. The smart sprayer^®^ can be operated on maize, sugar beet and sunflower at a speed of 10 km h^−1^. Each of the 142 nozzles with a width of 25 cm on the 48 m boom is controlled by a camera. However, image analysis is less sophisticated than that of AI-based robots. In the first imaging process, crop rows are excluded from the images. Then, weed coverage is measured in the inter-row area. If weed coverage exceeds a variable threshold, the nozzle is turned on [8,9,10]. In transplanted vegetable crops, simple features such as size, area and distance between plants allow crop/weed classification (e.g., K.U.L.T.i-Select^®^, Garford Robocrop InRow Weeder^®^ and Steketee IC^®^). Transplanted crops are usually much bigger than the weeds. Non-imaging spot sprayers such as Trimble Weed Seeker^®^ (Westminster, CO, USA) detect green vegetation using spectrometers that detect the reflectance of red and near-infrared light. Since plants reflect more infrared light and less red light than soil, a simple ratio of IR/red can be used to differentiate between green plants and soil. Trimble Weed Seeker^®^ was developed for non-selective weed control in the inter-row area or for fallow applications [11]. If the exact position of each crop seed was recorded with RTK-GNSS, some robots, such as Farmdroid FD20^®^ (Vejen, Denmark), can guide the spot sprayer exactly on the top or between each crop plant.

UAV images cover a larger area than tractor-mounted sensors. With UAVs, one hectare of an arable field could be sampled in less than 15 min [12]. If precise weed application maps could be automatically generated from the UAV images, sampling costs for weed scouting could be significantly reduced, and the driving speed for spot spraying could be increased with UAV images. So far, mainly patches of perennial weed species such as *Cirsium arvense* (L.) Scop. and *Rumex crispus* L. and grass weeds have been classified and georeferenced with UAV systems for patch spraying, resulting in 26% herbicide savings compared to a broadcast application [12,13,14]. Classified maps were used for site-specific weed control with a hoe or sprayer. The spatial resolution of UAV images has, so far, been too low for single-weed plant detection [13,14,15].

The objective of this study was to develop and investigate a spot-spraying system for weed control in maize based on UAV images. The precise classification of single-weed plants into UAV images was tested. Weed control efficacy (WCE) of spot spraying based on classified spray maps was assessed for early and late post-emergent herbicide applications. The hypotheses tested were that (i) more than 90% of all weeds were classified in the UAV images, (ii) WCE and maize yield of spot spraying were equal to the broadcast herbicide application and (iii) early post-emergence spot spraying would allow higher herbicide savings than late applications.

## 2. Materials and Methods

### 2.1. Experimental Sites

In 2023, four on-farm field experiments in maize (*Zea mays* L.) were conducted. The experiments were located in Southwestern Germany, two in Schlierbach (48.7° N, 9.5° E) at an elevation of 355 m above sea level and two in Bad Boll (48.6° N, 9.6° E) at an elevation of 411 m above sea level. The soil texture was classified as a clayey loam. The average yearly temperature in this region in 2023 was 11.7 °C, which was 1.1 °C higher than the long-term average. The average yearly rainfall in 2023 was 923 mm, which is 85 mm higher than the long-term average. Cultivars and dates of treatments are presented in Table 1.

BadBoll_1 was cultivated twice with a disc harrow approximately 10 cm deep (Amazone Catros, Hasbergen, Germany) and once with a rotary harrow (Kuhn HRB 503, Landau, Germany) before sowing. BadBoll_2 was ploughed 25 cm deep (Kuhn Vari-Master 153 6 T, Landau, Germany) in February. Three days before sowing the maize, the field was cultivated twice with a rotary harrow approximately 15 cm deep. In both fields, maize was sown with a seed density of 8.5 seeds m^−2^ and a row distance of 75 cm. Schlierbach_1 and Schlierbach_2 were ploughed in February 25 cm deep (Kuhn Multi-Master 152, Landau, Germany) and cultivated twice with a disc harrow (Kerner Corona 250, Aislingen, Germany) and once with a rotary harrow (Kuhn HRB 503, Landau, Germany) before sowing. On both fields, maize was sown with 9 seeds m^−2^ and a row distance of 65 cm. Maize was sown with a single grain seeder (Amazone Precea, 6 m, Hasbergen, Germany).

### 2.2. Experimental Design and Treatments

All four trials were set up in a randomized complete block design with four repetitions. Each of the four experiments contained 7 treatments. Post-emergence herbicides were applied early (Post Early), late (Post Late) and early after a pre-emergence herbicide treatment (Pre + Post). All three treatments were realized as a broadcast treatment across the whole plot (B) and as a spotting treatment for the post-emergence herbicides (S). A control treatment was included in all experiments without any weed control method (CON). For pre-emergence application, 3.5 L ha^−1^ Stomp Aqua^®^ (455 g L^−1^ pendimethalin, EC, BASF) and 1.0 L ha^−1^ Spectrum^®^ (720 g L^−1^ dimethenamid-P, EC, BASF) were used. Post-emergence treatments were realized with 1.5 L ha^−1^ MaisTer Power^®^ (30 g L^−1^ foramsulfuron + 9.77 g L^−1^ thiencarbazone + 0.85 g L^−1^ iodosulfuron + 15 g L^−1^ cyprosulfamide, SC, Bayer CropScience (Leverkusen, Germany)). The plot size in each experiment was 3 m × 10 m, with the longer side in the sowing direction. Herbicides were sprayed with an Amazone UF 2002 (Amazone, Hasbergen, Germany) with a width of 21 m (Figure 1) and single nozzle control. The sprayer was calibrated for a volume of 220 l ha^−1^ water at a pressure of 2.5 bar and a speed of 6 km h^−1^. The boom had a height of 0.5 m above the canopy. Injector compact broadcast spray nozzles IDK 90–04 C were used (Lechler, Metzingen, Germany).

### 2.3. UAV Imaging and Weed Detection

Aerial images were taken with a Quadrokopter MK-U25 (Multikopter.de, Sankt Katharinen, Germany) equipped with a multi-spectral camera system (SAMCAM1, 6 × 16 Mpic multihead camera system, SAM-DIMENSION, Stuttgart, Germany) and an RTK-GNSS (Figure 2). Only RGB images were used for this study. The camera had a global shutter and industrial optics (Zeiss, Jena, Germany). Images were taken from a height of 50 m, resulting in a ground sampling resolution of 1.4 mm per pixel. The drone flew at a speed of 40 km h^−1^, resulting in a performance of 57 ha h^−1^. Images were automatically stitched and georeferenced to an Ortho-Image for each experiment using SAM Dimension UAV image analysis. After segmentation using the Excessive Green Index, the Ortho-images were classified using the trained neural network MobileNetV2 [16] from SAM Dimension with ‘maize’, ‘weed’ and ‘background’ as output variables. Maize and weeds were trained from the one-eighth leaf growth stages. Boxes of 1 m × 1 m were created around each classified weed and marked in the spray map. Due to the low competitive ability of maize to suppress weeds [2], a zero weed threshold was used for the application maps. Drone images and spray maps were created 2–4 days before spot spraying on 12 June 2023 and 20 June 2023. Spray maps were uploaded on the Amatron4 spray terminal (Amazone, Hasbergen, Germany) for spot spraying (Figure 3 right).

For manual weed detection, UAV images were loaded into the geographic information system ArcGIS (ESRI, Redlands, CA, USA). Images were then overlaid with the plot map. The images were manually zoomed to a spatial resolution of 1.4 mm per pixel (Figure 3 left). If weeds were detected, they were marked with their coordinates and later overlaid with the automatically classified map (Figure 3 middle). With this approach, the accuracy of CNN-based weed classification was approved.

Herbicide savings were recorded for each spotting treatment by calculating the spray volume applied and relating this number to the spray volume of a broadcast application.

### 2.4. Assessments and Data Analysis

The weed density by species (plants m^−2^) was assessed before and 14 days after post-emergence herbicide application. Weeds were counted at six randomly selected positions of each plot using a 0.1 m^2^ frame. Maize plants were counted four times with a meter stick placed along the crop row.

Weed control efficacy (WCE) was calculated according to Equation (1).
WCE = 100 (1 − Wa/Wb)(1)
where Wa represents the weed density 14 days after application, and Wb is the density of weeds before application. Crop stand loss (CL) was calculated according to Equation (2).
CL = 100 (1 − La/Lb)(2)
where La represents the crop density 14 days after application, and Lb is the crop density before application.

Silage maize yield was assessed by harvesting the two center rows of each plot on 20 September 2023 and 25 September 2023 with a Baural SF 2000 (Zürn, Germany), when maize had approximately 40% dry matter content. The dry matter of the maize was determined by drying the chopped plants at 80 °C for 48 h in a drying chamber.

The accuracy of UAV weed maps was measured by comparing weed plants in the manually classified map with the weeds in the automatically classified maps. Similar to previous studies [8,11], a threshold of 0 weeds m^−2^ was applied to create a spray map from the weed map.

Data were analyzed using the statistical software R Studio (Version 4.1.1, RStudio Team, Boston, MA, USA). The residuals of the data set were tested for variance homogeneity and normal distribution prior to the analysis, which was given in all data sets. The different treatments were assigned as fixed effects, and the different locations were assigned as random effects. Prior to the analysis, outliers were removed to improve the model fit. To analyze the data, the mean values of the four experimental plots were pooled in order to carry out a one-factor analysis of variance. This kind of analysis was possible because there were no significant effects between the experiments. An analysis of variance (ANOVA) was carried out, and Tukey’s honest significant difference (HSD) test (α ≤ 0.05) was performed as a pairwise comparison of means. The model used for the analysis was a linear mixed model (LME):y_*ik*_ = *µ* + ω*k* + *a*_*i*_ + u_*k*_ + *e*_*ik*_(3)
where y*_ik_* is the result (e.g., WCE) of treatment *i* in block *k*, *µ* is the general mean, ω*k* is the fixed block effect, *a*_*i*_ is the fixed effect of the treatment, u_*k*_ are the random effects of the locations, and *e*_*i**k*_ is the residual error of the plot.

## 3. Results

### 3.1. Accuracy of the Weed Classification in UAV Images

On average, 94% of all weed plants were correctly identified in the UAV images using the automatic AI classifier. In previous years, the automatic classifier has been trained with many UAV images containing weeds and maize. Only 1.5% of the weeds were not detected by the automatic classifier but were manually detected in the drone images. The automatic classifier misclassified 4.5% of objects as weeds, which were manually identified as soil/mulch in the drone images. This means that herbicides were applied to 4.5% of the field where no weeds were present. Those misclassifications resulted from reflecting soil particles such as mulch and stones.

### 3.2. Weed and Crop Densities

*Poa annua* L. (78.4% frequency), *Chenopodium album* L. (10.5%), *Polygonum persicaria* Gray (4.2%), *Echinocloa crus-galli* (L.) P. Beauv (2.9%) and *Cirsium arvense* (L.) Scop. (1.3%) were the most abundant weed species in the experiments. Weeds were in the 2–4-leaf stage at the time of sampling 20 days after sowing (DAS). Maize density was not affected by herbicide treatments. On average, eight maize plants m^−2^ were counted in each treatment and experiment. Less than 5% of crop losses were observed in the treatments.

Weed density 20 DAS before early post-emergence herbicide application and spot spraying were considerably low, ranging between 3 plants m^−2^ (B–Post Early) and 8 plants m^−2^ (S–Post Late) (Figure 4). There were no significant differences between the treatments. After 40 mm of rainfall on 18 June 2023, a flash of new emerging weeds was recorded before late post-emergence weed control (B-Post Late, S-Post Late), and the untreated control plots increased weed density in the three treatments to 52 plants m^−2^.

### 3.3. Weed Control Efficacy

Early spotting (S–Post Early) achieved statistically equal WCE with 86% as the early broadcast post-emergence herbicide application (B–Post early) with 76%. The combination of pre-emergence and early post-emergence spraying resulted even in significantly lower WCE (63%) than S–Post Early and B–Post Early. Pre-emergence herbicide applications did not increase WCE. Late spotting (S–Post Late) was significantly less effective (20%) than late broadcast post-emergence spraying (B–Post Late), probably because many weed seedlings that had emerged after heavy rainfall on 18 June 2023 were not detected in the drone images because maize leaves covered those weed seedlings. Therefore, they were not sprayed in the late spotting application (Figure 5).

### 3.4. Herbicide Savings of Spot Spraying

Early spotting resulted in 47% herbicide savings. Taking the misclassifications of the UAV images into account, 50% savings could have been realized with the approach in this study. The combined treatment B-Pre + S–Post Early) amounted to 44% herbicide savings (0% for the pre-emergence herbicide and 48% for post-emergence herbicide). The savings from late spotting (S–Post Late) were significantly lower, with only 12%, which was probably due to higher weed density at late post-emergence herbicide application (Figure 6).

### 3.5. Silage Maize Yield

Silage maize yield was not significantly affected by weed control treatments (Figure 7). The average yield ranged between 44.6 (CON) and 50.7 t ha^−1^ (B–Pre + B–Post Early). Maize was harvested with 42–45% dry matter content.

## 4. Discussion

A new spot-spraying system for weed control in maize based on UAV images is presented in this study. Compared to previous patch-spraying systems based on drone images against perennial weeds and grass weeds [12,13,14,15,17], spatial image resolution in the present study was much higher, so spraying was targeted on single weeds. Weeds were automatically classified with an accuracy of 94% in the UAV images with the neural network algorithm compared to manual weed classification in the digital images. This is consistent with [17], who achieved 92% accuracy in grass weed classification in drone images using a high-resolution RGB camera. Very few weed plants were not detected before early spot spraying in the UAV images. Therefore, the WCE of early spotting was very high (86%) comparing weed densities before and after spot spraying, which was equal to the broadcast herbicide application. Savings of early spot spraying amounted to 47%. Those data were obtained by relating the spray rate of spot spraying to the spray rate of the broadcast herbicide treatment. Savings could be further increased if application technology were improved or economic weed thresholds were applied. In the present study, conventional injector compact broadcast spray nozzles with a distance of 0.5 m were mounted on the boom. The nozzles overlap 0.25 m to each side, resulting in a spray width of 1 m. Due to the reaction time of the nozzle, herbicides were applied 0.5 m before and behind each weed plant. Therefore, the smallest spot was 1 m × 1 m. Nozzles with smaller bands of 0.06–0.25 m were implemented in other spot spraying systems with cameras mounted in front of the spray boom [8,18]. It can be questioned whether injector compact broadcast spray nozzles, in general, are suitable for spot spraying. Those nozzles need to overlap to achieve a homogeneous lateral spray rate [8]. This means that for the present study, weed plants on the side of the spray window received a lower rate than weeds in the center of the spray window. That could partly explain the lower WCE of late spot spraying compared to late broadcast herbicide application. New nozzle types with homogeneous distribution have been developed [7,8,9].

Late spotting at the 6–8-leaf stage of maize completely failed with only 20% WCE and 12% herbicide savings. This shows that the success of UAV-based spot spraying depends not only on the spatial resolution of the UAV images but, most importantly, on crop coverage. Maize leaves covered most parts of the in-row area and some parts of the inter-row area before late UAV sensing and spotting. Therefore, many small weeds were not detected. Under those conditions, cameras mounted on the robots could still identify most of the weeds because they can be placed underneath the crop canopy or look diagonally into the crop canopy [14].

Although it was not possible to use classified weed maps for real-time spot spraying, as is realized in many commercial weeding robots [5,6,7,8,9], UAV images provide several benefits compared to tractor-mounted camera images. If UAV images were autonomously classified and transferred to the spray computer, weed scouting and spot spraying could be realized much faster than with real-time systems because the field of view and speed of UAV cameras is higher than for tractor-mounted systems. Spot spraying based on UAV images can be applied at higher speeds than spot spraying with cameras mounted on the vehicle because driving speed is not limited by the speed of image classification [18].

The classifier distinguished between crops and weeds. Weed species were not differentiated. The algorithm for plant species classification was very similar to the Smart Sprayer^®^ [8]. If neural networks were applied for plant species discrimination [9,19,20,21], spot spraying could be improved in many ways. For competitive weed species, lower weed control thresholds could be realized [3]. In addition to that, grass weeds and broadleaved weed species could be treated with different herbicides if direct injection systems or multi-tank sprayers were used [2].

Pre-emergence herbicide applications, in combination with post-emergence herbicides, in the present study did not provide higher WCE than post-emergence applications alone. This can be explained by low soil moisture conditions at the time of application, which inhibits the activation of pre-emergence herbicides [22]. Spotting is mostly realized with post-emergence herbicides [18]. Pre-emergence herbicides can be spotted in combination with RTK-GNSS-guided seeding technologies, e.g., Farmdroid FD20^®^ [18]. In the present study, early spotting of post-emergence herbicides resulted in 86% WCE, which is a clear indicator that post-emergence herbicides play a major role in chemical weed control.

Weeds in the present study did not significantly affect yield. Early weed competition was probably too low to cause a significant yield loss in the untreated control [23]. Late-emerging weeds probably could not compete with the maize plants, so they did not affect the yield. In general, maize is very sensitive to early weed competition. Therefore, spot spraying significantly increased the yield of sunflower and maize [8].

## 5. Conclusions

UAV-based weed mapping, in combination with spot spraying in maize, offers a robust and precise alternative to robotic weeding with sensors mounted on the tractor. In the present study, a high-resolution UAV camera was used that was capable of detecting single annual weed plants. However, weed classification in the UAV images was only possible until the four-leaf stage of maize. Later, maize shaded the late merging weeds on the ground so that they were not detected. With up to 86% WCE and 47% herbicide savings, early spot spraying represents a great improvement compared to broadcast herbicide treatments and allows for a significant reduction of herbicide use without reducing WCE. This study encourages the further development of UAV-based weed classification in other crops, such as sugar beet and sunflower. Compared to tractor-mounted camera systems, UAV-based weed classification can reduce costs for weed sampling.

## Figures and Tables

**Figure 1 plants-13-02164-f001:**
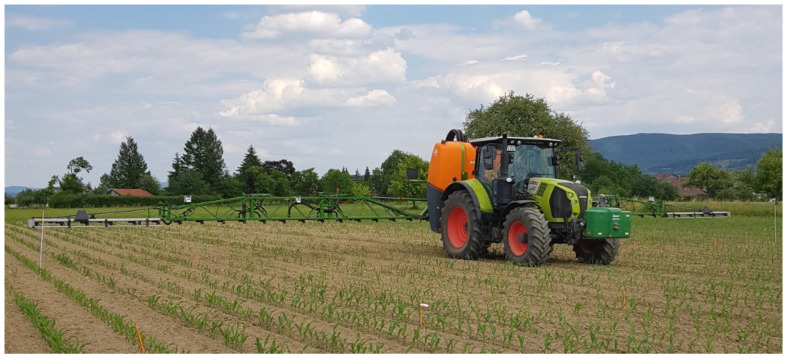
Amazone UF2002 sprayer with a spry boom of 21 m width.

**Figure 2 plants-13-02164-f002:**
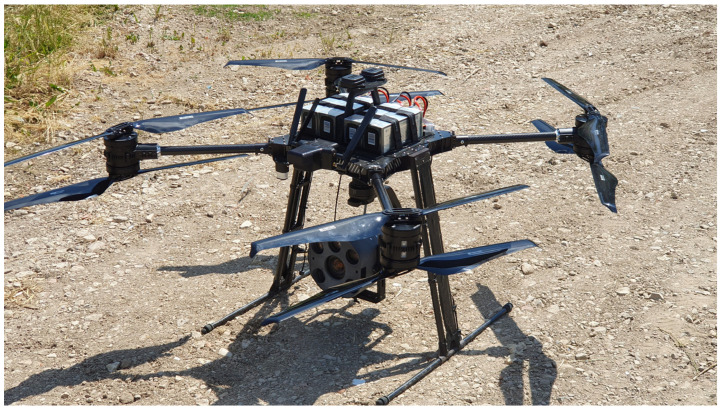
Quadrokopter MK-U25 from multikopter.de (Sankt Katharinen) and SAM Dimension (Stuttgart), Germany.

**Figure 3 plants-13-02164-f003:**
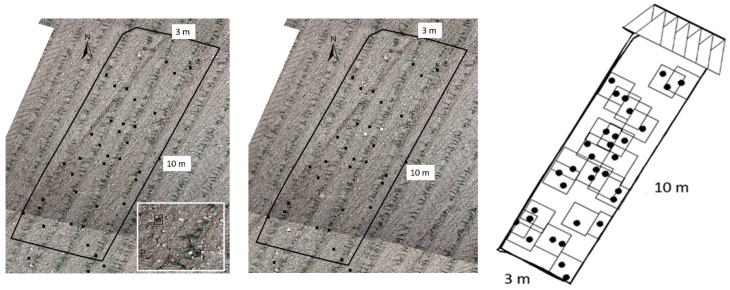
Drone map of an early post-emergence spot spraying plot in Bad Boll_1 with the weeds manually marked in black (**left**). Drone map with the weeds automatically correctly classified (black); white dots indicate weeds that were not detected by the image classifier; gray dots represent weeds that were misclassified by image classifier (**middle**). Application map of the spot sprayer (frames with the weed spots were applied (**right**).

**Figure 4 plants-13-02164-f004:**
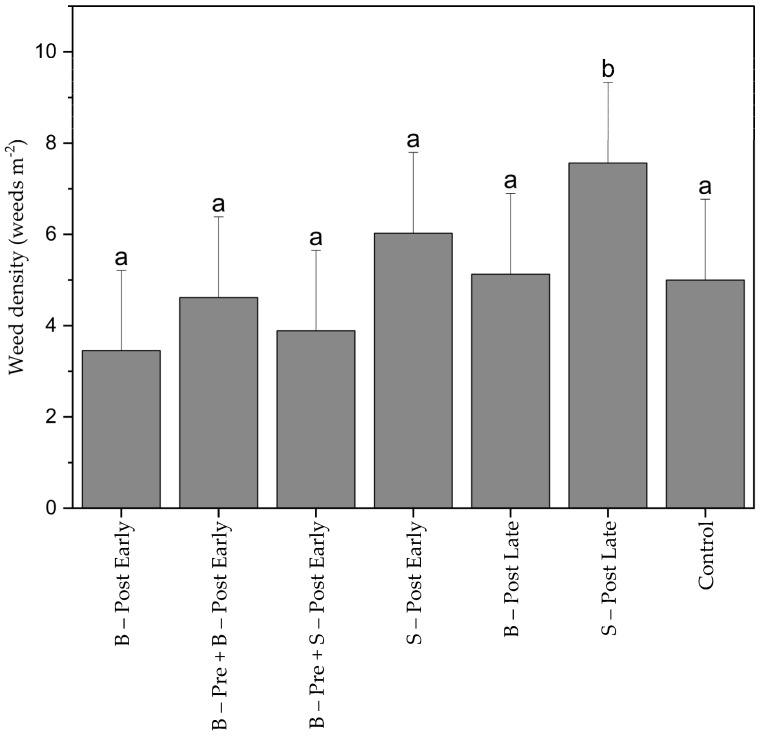
Average weed density (weeds m^−2^) in maize counted 20 days after sowing (DAS). B–Post Early = early broadcast post-emergence treatment, B–Pre + B–Post Early = broadcast pre-emergence treatment combined with early broadcast post-emergence treatment, B–Pre + S–Post Early = broadcast pre-emergence treatment combined with early post-emergence spot spraying, S–Post Early = early post-emergence spot spraying, B–Post Late = late broadcast post-emergence herbicide treatment, S–Post Late = late post-emergence spot spraying, Control = untreated control. Means with the same letter are not significantly different according to HSD-test (α ≤ 0.05). Bars represent the standard error (SE) of the mean.

**Figure 5 plants-13-02164-f005:**
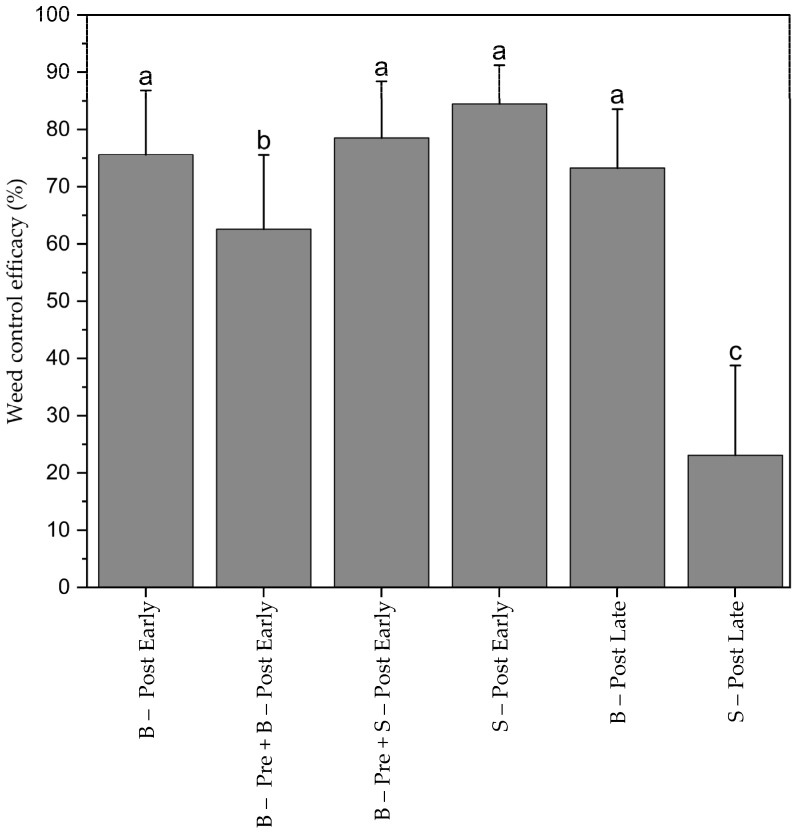
Weed control efficacy (%) for all treatments. Means with the same letter are not significantly different according to HSD-test (α ≤ 0.05). Bars represent the standard error (SE) of the mean.

**Figure 6 plants-13-02164-f006:**
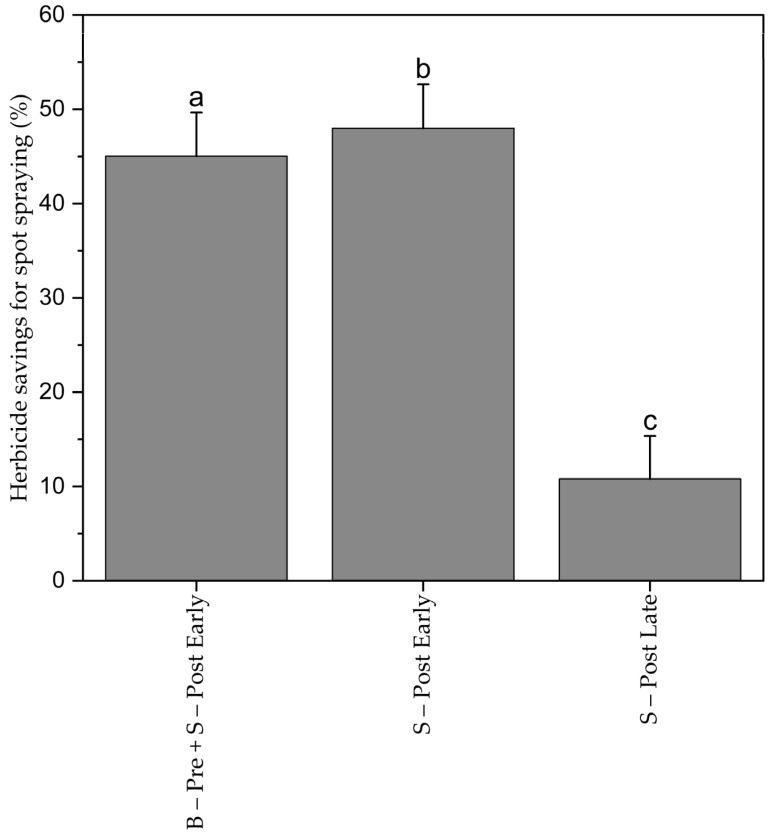
Herbicide savings for spot-spraying treatments. Means with the same letter are not significantly different according to HSD test (α ≤ 0.05). Bars represent the standard error (SE) of the mean.

**Figure 7 plants-13-02164-f007:**
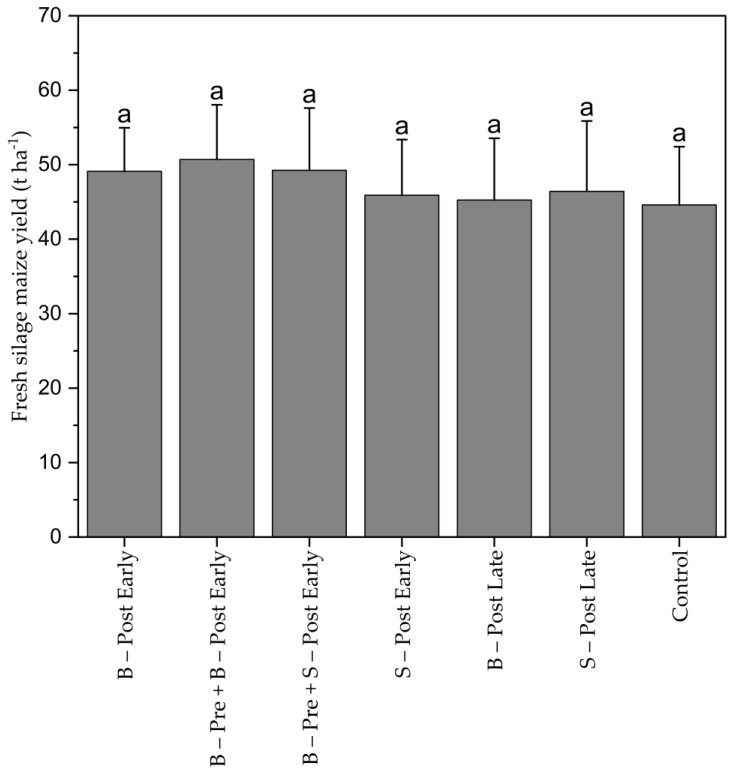
Fresh silage maize yield (t ha^−1^) harvested from the two center rows of each plot. Means with the same letter are not significantly different according to HSD test (α ≤ 0.05). Bars represent the standard error (SE) of the mean.

**Table 1 plants-13-02164-t001:** Maize cultivars and dates (MM-DD-YY) of sowing and herbicide application (spot spraying and broadcast application).

Experiment	Cultivar	Sowing Date	Pre-Emergence	Early Post-Emergence (2–4-Leaf Stage)	Late Post-Emergence (BBCH 6–8-Leaf Stage)
Bad Boll_1	Limagrain LG631.229	25 May 2023	1 June 2023	16 June 2023	24 June 2023
Bad Boll_2	Pioneer 8255	30 May 2023	1 June 2023	16 June 2023	24 June 2023
Schlierbach_1	SY Glorius	18 May 2023	24 May 2023	15 June 2023	24 June 2023
Schlierbach_2	Pioneer P9757	18 May 2023	24 May 2023	14 June 2023	24 June 2023

## Data Availability

Data are made available by the authors upon request.

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
