# Peer review of "Agronomic and Technical Evaluation of Herbicide Spot Spraying in Maize Based on High-Resolution Aerial Weed Maps—An On-Farm Trial"

_plants, 2024, doi:10.3390/plants13152164_

Round 1

Reviewer 1 Report

Comments and Suggestions for Authors

1 The structure of the manuscript needs to be adjusted, that is, from “Introduction”, “Materials and methods”, then “Results and discussion”, “Conclusions”.

2 Why listed the "robotic weeding" in keyword list? Actually the manuscript does not have detailed information for the robotic weeding, it might be the potential application. Please delete “robotic weeding”and “precision farming”in the keywords list, then add more related words to the list, such as,“spot spraying”and “weeds control efficacy”.

3 Please state more detailed information on how to form the prescription map and control the spotting spraying procedures in section“Materials and methods”.

4 There are several wrong words in the whole manuscript, such as, “threshol” in line 46, “manally” in line 80, “clasified” in line 262, “than” and “overlad” in line 267, “ist” in line 318, ....... Please check the whole manuscript to further find and correct all the wrong words (including some wrong punctuation symbols).

5 You may move the contents of sub-section“2.1. Accuracy of the weed classification in UAV images” to section "3. Discussion" as there are some repetitive contents, or re-organize a section“Results and discussion”.

6 After the first annotation of an abbreviation, it can be used directly afterwards because of no need for repeated annotations, such as, “20 days after sowing (DAS)”.

7 There are detailed figure notes in Figure 1, and you don’t need to give the repeated figure notes for the same things in Figure 2 and afterwards. Then please check the word“early” in all the figures and in the whole manuscript, like “S–Post Early”and “S–Post”.

8 All the values of percentages in section “3. Discussion” should be explained clearly that you obtained the values through what kind of measures.

9 Please mention the type and manufacturer of "flat jet nozzles" in lines 162-163 and other places. And what is the particular model type of "Injector compact broadcast spray nozzles" in lines 247-248?

10 Can you fill in the related contents of lines 221-231 into Table 1 to improve the readability?

11 Please mark clearly the dimension lines of 3m and 10m in Figure 7 (right).

12 Why“a threshold of 0 weeds m-2”in line 302? Usually below a certain threshold is sufficient for weeds control, and it is not necessary to remove all weeds. Please give a reasonable explanation for “0”.

13 You may give a little bit detailed information for the section "5. Conclusions", such as summarizing the main research contents and the main results.

Comments on the Quality of English Language

Moderate editing of English language required. Please check the related comments in “Comments and Suggestions for Authors
”. 

Author Response

Point-by-point-response to both reviewers.

On behalf of the co-authors I would like to thank both of you for the fair and helpful review of our manuscript. We responded to reviewer 1 below and to reviewer 2 in the attached PDF. You find our revisions in the manuscript highlighted in red color. The figures were all revised as suggested.

Reviewer 1:

  1. The structure of the manuscript needs to be adjusted, that is, from “Introduction”, “Materials and methods”, then “Results and discussion”, “Conclusions”. We agree and changed accordingly.

2 Why listed the "robotic weeding" in keyword list? Actually the manuscript does not have detailed information for the robotic weeding, it might be the potential application. Please delete “robotic weeding”and “precision farming”in the keywords list, then add more related words to the list, such as,“spot spraying”and “weeds control efficacy”. We agree and changed accordingly.

3 Please state more detailed information on how to form the prescription map and control the spotting spraying procedures in section“Materials and methods”. We agree and changed accordingly.

4 There are several wrong words in the whole manuscript, such as, “threshol” in line 46, “manally” in line 80, “clasified” in line 262, “than” and “overlad” in line 267, “ist” in line 318, ....... Please check the whole manuscript to further find and correct all the wrong words (including some wrong punctuation symbols). Thanks for highlighting those mistakes. We ckecked the text and corrected some more mistakes.

5 You may move the contents of sub-section“2.1. Accuracy of the weed classification in UAV images” to section "3. Discussion" as there are some repetitive contents, or re-organize a section“Results and discussion”. We restrutured the Results and Discussion parts.

6 After the first annotation of an abbreviation, it can be used directly afterwards because of no need for repeated annotations, such as, “20 days after sowing (DAS)”. We agree and changed accordingly.

7 There are detailed figure notes in Figure 1, and you don’t need to give the repeated figure notes for the same things in Figure 2 and afterwards. Then please check the word“early” in all the figures and in the whole manuscript, like “S–Post Early”and “S–Post”. We agree and changed accordingly.

8 All the values of percentages in section “3. Discussion” should be explained clearly that you obtained the values through what kind of measures. We agree and changed accordingly.

9 Please mention the type and manufacturer of "flat jet nozzles" in lines 162-163 and other places. And what is the particular model type of "Injector compact broadcast spray nozzles" in lines 247-248? We agree and changed accordingly.

10 Can you fill in the related contents of lines 221-231 into Table 1 to improve the readability? We tried this but had to realize that this would make Table 1 difficult to read.

11 Please mark clearly the dimension lines of 3m and 10m in Figure 7 (right). We agree and changed accordingly.

12 Why“a threshold of 0 weeds m-2”in line 302? Usually below a certain threshold is sufficient for weeds control, and it is not necessary to remove all weeds. Please give a reasonable explanation for “0”. We explained in the text.

13 You may give a little bit detailed information for the section "5. Conclusions", such as summarizing the main research contents and the main results. We agree and changed accordingly.

Reviewer 2 Report

Comments and Suggestions for Authors

Dear Authors

All comments, remarks and questions are included in the manuscript 

Author Response

Dear reviewer 2,

thanks for your valuable comments and suggestions. We replied to your comments in the PDF and revised the manuscript accordingly.

Best regards

Roland Gerhards

Round 2

Reviewer 1 Report

Comments and Suggestions for Authors

“3. Discussion”  in line 260 should be “4. Discussion”.

Reviewer 2 Report

Comments and Suggestions for Authors

Dear Authors

Thank you for taking into account the comments, remarks and for answering the questions